# Bacterial Metabolic Potential in Response to Climate Warming Alters the Decomposition Process of Aquatic Plant Litter—In Shallow Lake Mesocosms

**DOI:** 10.3390/microorganisms10071327

**Published:** 2022-06-30

**Authors:** Penglan Shi, Huan Wang, Mingjun Feng, Haowu Cheng, Qian Yang, Yifeng Yan, Jun Xu, Min Zhang

**Affiliations:** 1Hubei Provincial Engineering Laboratory for Pond Aquaculture, Engineering Research Center of Green Development for Conventional Aquatic Biological Industry in the Yangtze River Economic Belt, Ministry of Education, College of Fisheries, Huazhong Agricultural University, Wuhan 430070, China; spl100306@163.com (P.S.); 17671718581@163.com (M.F.); chenghw3829@163.com (H.C.); yang123321@foxmail.com (Q.Y.); yanyifeng2@163.com (Y.Y.); 2State Key Laboratory of Marine Resource Utilization in South China Sea, Hainan University, Haikou 570228, China; wanghuan@ihb.ac.cn; 3Donghu Experimental Station of Lake Ecosystems, State Key Laboratory of Freshwater Ecology and Biotechnology of China, Institute of Hydrobiology, Chinese Academy of Sciences, Wuhan 430072, China; xujun@ihb.ac.cn

**Keywords:** climate warming, bacteria, FAPROTAX function prediction, litter, mesocosm, *Potamogeton crispus* L.

## Abstract

Increased decomposition rates in shallow lakes with global warming might increase the release of atmospheric greenhouse gases, thereby producing positive feedback for global warming. However, how climate warming affects litter decomposition is still unclear in lake ecosystems. Here, we tested the effects of constant and variable warming on the bacterial metabolic potential of typically submerged macrophyte (*Potamogeton crispus* L.) litters during decomposition in 18 mesocosms (2500 L each). The results showed that warming reduced main chemoheterotrophic metabolic potential but promoted methylotrophy metabolism, which means that further warming may alter methane-cycling microbial metabolism. The nitrate reduction function was inhibited under warming treatments, and nitrogen fixation capability significantly increased under variable warming in summer. The changes in dissolved oxygen (DO), pH, conductivity and ammonium nitrogen driven by warming are the main environmental factors affecting the bacteria’s metabolic potential. The effects of warming and environmental factors on fermentation, nitrate reduction and ammonification capabilities in stem and leaf litter were different, and the bacterial potential in the stem litter were more strongly responsive to environmental factors. These findings suggest that warming may considerably alter bacterial metabolic potential in macrophyte litter, contributing to long-term positive feedback between the C and N cycle and climate.

## 1. Introduction

The increase in average temperature and frequent occurrence of extreme weather caused by climate change have a more and more serious impact on shallow lake ecosystems [1,2]. Macrophytes such as submerged plants are key components of shallow lake ecosystems and maintain the basic functions and biodiversity of lake ecosystems [3]. However, due to multiple anthropogenic pressures such as climate change, eutrophication and aquaculture, the abundance of submerged plants in lakes has decreased [4]. *Potamogeton crispus* L. is a submerged aquatic macrophyte native to Europe and Asia that occurs in many freshwater ecosystems throughout China [5]. Xu et al. (2020) [6] found that warming accelerated the growth and senescence of *P. crispus*, suggesting a more important role in maintaining the clear water state in winter–early spring but concomitant to possible earlier turbid states in summer [3,6]. The decomposition and recycling of *P. crispus* are also key processes in the aquatic ecosystem, linking nutrient cycle and energy transfer between water and sediment [7].

Decomposition is the basic process that regulates the carbon cycle [8], driven by both microorganisms and invertebrates [9]. At the same time, it will be affected by climate change, the loss of species, and other external factors [10]. Temperature is one of the most important factors affecting decomposition [11]. The effect of temperature on the biomass of decomposers will lead to the change of decomposition rate of litter [12,13]. Previous studies found that temperature was positively correlated with decomposition rate, and the positive effects of water temperature gradients (5–15 ℃) on leaf decomposition have been widely reported [14,15]. Warming in the cold season stimulated the decomposition of litter more severely [16]. For example, Mao et al. (2018) [17] describe microbial respiration and litter decomposition rates increasing with temperature in peatlands of north-eastern China. Warming promoted the growth and reproduction of microorganisms and increased the leaching of soluble compounds [18]. The enzyme activity of microbial decomposers was stimulated by climate warming, and the decomposition of litter was enhanced [19]. In addition to the increase in average temperature, the decomposition of litter under heat wave not only increased the decomposition rate, but also significantly increased bacterial diversity, and stimulated the heterotrophic processes more than autotrophic processes, thereby increasing the rates of detrital decomposition and ecosystem respiration [20,21]. Relative to leaf litter in the heatwaves treatments, greater increases in processing rates for wood were observed [20], suggesting that enzyme activity related to episodic temperature increases may be coupled with more recalcitrant organic matter sources [12,22]. Seasonal factors should not be ignored in the studies of the decomposition process of litter. The various effects of seasonality and differences in population and community structure that represent proxies for long-term shifts can obscure the true, inherent temperature dependence of detrital processing [23]. Therefore, it is important to explore the changes of microbial metabolic function with seasonal temperature increase.

In addition to the direct impact of climate warming on the decomposition of litter, changes in environmental factors driven by climate warming may indirectly affect the metabolic process of litter [23,24,25,26]. In many aquatic systems, low concentrations of nitrogen and especially phosphorus limit the metabolic rate of bacterial communities [27]. In oligotrophic waters, it was found that substrate utilization mainly regulated bacterial growth and respiration [28]. However, temperature explained 47% of bacterial growth efficiency and 46% of bacterial respiration when nutrients were not restricted [25]. Anoxic conditions may foster the decay of the litter, rendering litter more vulnerable to bacterial decomposition [24]. In the study of the aquatic ecology model of tank-forming bromeliads, it was found that pH strongly induced the changes of abundance of microbial community composition in bromeliad basin water [26]. Roots with high fiber content in litter tissue decompose more slowly than other tissues (leaves and petioles) [29]. The influence of a single environmental change on the degradation process of litter is not convincing enough. In the actual environment, it is usually the result of a combination of biological and abiotic factors.

Previous studies have been reported on temperature driving litter decomposition and influencing microbial community structure in terrestrial and aquatic ecosystems [21,30,31]. For example, warming induced the expansion of shrubs with recalcitrant leaf litter across cold biomes [30]. However, whether warming amplifies the effects on microbial metabolic potential during litter decomposition is unknown. Here, we experimentally tested the effects of three temperature scenarios (ambient temperature, constant and variable warming) on ecologically relevant functional groups of bacterial communities of different tissues litter during decomposition. We specifically hypothesized that (1) differences in temperature due to season and warming will induce shifts in the bacterial functional structure of leaf and stem litter decomposition and further alter bacterial metabolic pathways at the stage of litter decomposition through the change of water physical–chemical factors; (2) the responses of bacterial metabolism potential to warming and environmental factors differ between leaf and stem litter.

## 2. Materials and Methods

### 2.1. Mesocosm Experiment Set Up and Management

The experiment was carried out in 18 polyethylene tanks, each 1.5 m in diameter and containing about 2500 L of water with a depth of about 1.4 m, located in the undergraduate experimental teaching base of Huazhong Agricultural University at Wuhan City (30°28′15″ N, 114°22′17″ E). All experimental mesocosms received 10 cm of mixed lake sediment as well as 1 m depth of lake water at the beginning of the experiments. Lake sediment was collected with a Peterson grab sampler from a pelagic area in Lake Liangzihu (30°11′3″ N, 114°37′59″ E) in late October 2018. Lake Liangzihu is a mesotrophic shallow lake in the middle and lower reaches of the Yangtze River basin, with TN and TP concentrations in the water column of our sampling area of approximately 0.432 mg L^−1^ and 0.023 mg L^−1^, respectively [6,32].

The sediments were thoroughly mixed in a clean container before immediate transfer to the experimental mesocosms. All water added to the experimental mesocosms was filtered through a 20 mm mesh. We fixed temperature sensors (DS18B20, Maxim IC, Dallas, TX, USA) and heating units (A10-3, Xin Shao Guangzhou, China) to each polyethylene mesocosms to monitor the temperature of all treatments in real time [33]. All mesocosms were left for colonization under ambient conditions for two months (November–December 2018) before the start of the experiment.

These mesocosms mimic shallow lake systems under three temperature scenarios with six replicates: (1) ambient temperature (C), (2) constant warming (T, +4 ℃ above ambient temperature) according to IPCC climate scenario RCP8.5 [34], and (3) variable warming (V, with water temperatures fluctuating 0~8 ℃ relative to constant warming) [35]. The control and two treatments were randomly assigned to the 18 mesocosms. The experiment ran from 1 January 2019 to 30 October 2019.

### 2.2. Sampling and Chemical Analyses

DO, pH and conductivity were measured monthly with HACH HQD Portable Meters (HQ40d, HACH, Loveland, CO, USA). Depth-integrated water samples were collected monthly using a transparent plexiglas tube (diameter 70 mm, length 1 m) to analyze total nitrogen (TN), total phosphorus (TP) concentrations and chlorophyll a (Chl-a) concentrations. TN and TP were determined by spectrophotometry (UV-2800, Unico, Shanghai, China) after digestion with alkaline potassium persulfate [36]. Chl-a was determined by water filtration on Whatman GF/C filters and spectrophotometric (UV-2800, Unico, Shanghai, China) analysis after ethanol extraction [37]. Water was filtered through GF/C filters to determine the NH_4_^+^-N, NO_3_^−^-N and PO_4_^3−^-P concentrations. The content of NH_4_^+^-N in the water column was determined by Nessler’s reagent colorimetric method, and NO_3_^−^-N content was determined by ultraviolet spectrophotometry method [36]. The concentration of PO_4_^3−^-P in the water column was determined using the molybdenum blue method [38]. 

### 2.3. Decomposition Experiment Set Up and Sampling

To measure the decomposition of the aquatic macrophytes litter, a litter bag method decomposition experiment was conducted in the mesocosms [39,40]. We collected *Potamogeton crispus* L. from Lake Liangzihu in January 2019 before the start of the decomposition experiment. Firstly, we separated the leaves and stems of the *Potamogeton crispus* L. shipped back to the laboratory and dried them to constant weight at 60 °C. Then, the leaf and stem litter were placed in polyethylene litter bags (10  ×  6 cm^2^) with a mesh size of 425 μm. To ensure comparability between consecutive samples, the litter samples in all the litter bags were standardized to the same weight (0.50 ± 0.01 g) using an analytical balance (AE224, SOPTOP, China). Before the experiment, we numbered all litter bags and recorded them. We hung all these litter bags above the sediment in the mesocosms on 1 February, 1 May and 1 August 2019, respectively. The experiment lasted for a total of 60 days, and litter bags were collected from the control and the warming treatments on the 30th day after hanging the litter bags. We collected bacteria by swabbing the entire surface of the litter with sterile cotton swabs and placing them in a 15 mL centrifuge tube [41]. A total of 54 bacteria samples were collected in spring, summer and autumn and stored in a freezer at −80 °C until DNA was extracted.

### 2.4. DNA Extraction, PCR Amplification

DNA was extracted from 54 bacterial samples according to the instructions of the FastDNA^®^ Spin Kit for Soil (MP, Biomedicals, Santa Ana, CA, USA). DNA concentration and purity were detected using a NanoDrop 2000 (Thermo Fisher Scientific, Wilmington, DE, USA), and DNA quality was checked by 1% agarose gel electrophoresis. The hypervariable region V3–V4 of the bacterial 16S rRNA gene were amplified with primer pairs 338F (5′-ACTCCTACGGGAGGCAGCAG-3′) and 806R (5′-GGACTACHVGGGTWTCTAAT-3′) by an ABI GenAmp^®^ 9700 PCR thermocycler (ABI, Foster City, CA, USA) [42]. The reaction mixture contained 10 ng of genomic DNA and 0.2 μL of BSA solution as a template, 4 μL of Ex Taq™ buffer (5×), 2 μL (2.5 mM) of dNTP mix, 5 μM of each primer, and the final reaction volume of 20 μL. The cycling conditions were denaturation at 95 °C for 3 min, 27 cycles of denaturing at 95 °C for 30 s, annealing at 55 °C for 30 s and 45 s extension at 72 °C, single extension at 72 °C for 10 min, and end at 4 °C. Library construction, quality assessment, and sequencing (Illumina MiSeq PE300 platform (Illumina, San Diego, CA, USA)) were carried out following our previous article without any change [43].

### 2.5. FAPROTAX Analysis and Functional Prediction

The raw data sequences were processed following the course of quality trimming [44], paired-end sequence assembly [45], and chimera removal [46]. Operational taxonomic units (OTUs) were obtained with a 97% similarity level using Usearch (version 7.0 http://drive5.com/uparse, accessed on 8 October 2021)) [47]. The functional annotation of prokaryotic taxa (FAPROTAX) was a manually constructed functional annotation database that maps prokaryotic taxa (e.g., genera or species) to metabolic or other ecologically relevant functions (e.g., chemoheterotrophy, methylotrophy, or fermentation) based on the literature on culturable bacteria [48], which was used to predict the functional potential of microbial communities in litters under different treatments. The program includes a Python script for converting OTU tables into putative functional tables based on the taxa identified in a sample and their functional annotations in the FAPROTAX database. The complete database for FAPROTAX includes more than 7600 functional annotations from more than 80 functional groups, specifically adapted to the biogeochemical cycles of environmental samples (especially carbon, nitrogen, hydrogen, phosphorus and sulfur cycles) in marine and lake environments, in which carbon and nitrogen cycles are the bacterial metabolic cycles of our primary concern. Because of FAPROTAX, based on the published literature of culturable bacteria, it had better prediction accuracy. It is generally believed that predicting putative functional groups using this approach is superior to genomic prediction approaches based on sequence homology [48,49]. Our paired-end Illumina sequence data used in the summer are available in the Sequence Read Archive (SRA) of the National Center for Biotechnology Information (NCBI), BioProject accession number PRJNA705584. The paired-end Illumina sequence data in spring and autumn have not been published. The annotated OTU table of Greengenes or Silva database based on 16S was run through a Python script to match the species information in the FAPROTAX database (http://www.zoology.ubc.ca/louca/FAPROTAX/ (accessed on 7 April 2021)), and then the functional annotation prediction results of the microbial community were output.

The FAPROTAX table output 25 bacterial annotations related to carbon cycle and 14 bacterial annotations related to nitrogen cycle. However, the raw output of FAPROTAX presents multiple duplicities in the metabolic assignments. For example, the chemoheterotrophy metabolic function abundance includes aerobic chemoheterotrophy. For this reason, the FAPROTAX table has been debugged by eliminating duplicities in metabolic potentials in our research. In our work, the “aerobic chemoheterotrophy” by the FAPROTAX database will be referred to as “Chemo-1” and accounts for the main potential aerobic heterotrophic metabolic pathways. What is referred to in the FAPROTAX database as “chemoheterotrophy” has here been grouped as “Chemo-2”. “Chemo-2” only contains potential recalcitrant heterotrophic metabolic pathways, including lignin, chitin, xylan and cellulose, other than what we refer to as “Chemo-1”. The “methylotrophy” includes “methanol oxidation” and “methanotrophy”; we retain “methylotrophy”. The “aromatic compound degradation” adds to the group “aromatic hydrocarbon degradation”; we retain “aromatic compound degradation”. With regard to nitrogen, the “nitrite ammoniation” includes “nitrate ammonification”, and we retain “nitrite ammonification”, referred to as “ammonification”. Ammonification represents the reduction of dissimilated nitrate to ammonium (DNRA). Accordingly, the modified table without duplicity contains 10 bacterial annotations related to carbon cycle and 6 bacterial annotations related to nitrogen cycle (Appendix A).

### 2.6. Statistical Analysis

All statistical analyses were performed using R software (version 4.0.5, TUNA Team, Tsinghua University, Beijing, China), and graphs were made with the “ggplot2” R package. Water environment physicochemical properties and bacterial functional groups in *P. crispus* leaf and stem litter were analyzed using linear mixed models with the “lmer” functions from the “lme4” R package [50], and sampling date and tank number as a random effect for the models, respectively. Then, we performed post hoc pairwise comparisons among different treatments via Tukey’s test with the “emmeans” R package [51]. PCoA based on Bary-curtis distances were performed with the “vegen” R package. PERMANOVA analysis was performed by Vegan (R package) (version 2.5-7, Jari Oksanen Team, Helsinki, Finland) with the ‘Adonis’ function with 999 permutations. Before the analyses, variables were transformed using arcsine square root or log transformation as necessary. All of the analyses were performed at a 0.05 statistical significance level. 

## 3. Results

### 3.1. Conditions in the Experimental Mesocosms

During the experiment, the water temperature of treatments in the mesocosms followed the desired experimental design (Figure 1). The water temperature of constant warming treatment (T) was +4 ℃ higher than that of ambient temperature (C). In variable temperature treatment (V), the temperature fluctuates relative to constant warming treatment (T). Warming significantly enhanced conductivity and TN concentrations in all seasons (*p* < 0.05, Table 1). In summer, pH, DO and NO_3_^−^-N were significantly higher in the T treatment, and PO_4_^3−^-P contents were more pronounced by variable warming compared to the control (*p* < 0.05). 

### 3.2. Bacterial Functional Abundance Analysis

PCoA analysis showed that the clustering of bacterial functional groups under different seasons of warming treatments was significant (PERMANOVA, *p* < 0.001) (Appendix A). The average abundances of bacteria to carbon cycle were 16,704 ± 1577, 15,411 ± 1392 and 15,876 ± 12042 in CL (ambient temperature × leaf), TL (constant warming × leaf) and VL (variable warming × leaf) treatment, and 17,547 ± 1768, 17,294 ± 1693 and 18,765 ± 2279 in CS (ambient temperature × stem), TS (constant warming × stem) and vs. (variable warming × stem) treatment, respectively (Figure 2a). There was no significant difference in the abundance of bacterial functional groups related to carbon cycle in leaf and stem litter between treatments. The average abundance of nitrogen cycle bacteria was 1782 ± 333, 1830 ± 254 and 2527 ± 405 in CL, TL and VL treatment, and 6125 ± 1929, 2579 ± 418, 2738 ± 448 in CS, TS and vs. treatment, respectively. The abundance of bacterial functional groups related to nitrogen cycle in stems of the control treatment (C) was significantly higher than that in CL and TL (*p* < 0.05, Figure 2b).

### 3.3. Composition of Bacterial Functional Groups

The functional groups in all treatments of carbon cycle in leaf litter were Chemo-1 (48% ± 0.07), methylotrophy (11% ± 0.03), photoheterotrophy (11% ± 0.02), oxygenic photoautotrophy (10% ± 0.04), fermentation (10% ± 0.02), Chemo-2 (9% ± 0.02) and aromatic compound degradation (1% ± 0.00) (Figure 3a). The functional groups in all treatments of the carbon cycle in stem litter were Chemo-1 (46% ± 0.07), methylotrophy (9% ± 0.02), photoheterotrophy (10% ± 0.01), oxygenic photoautotrophy (10% ± 0.03), fermentation (10% ± 0.01), Chemo-2 (12% ± 0.04) and aromatic compound degradation (2% ± 0.01) (Figure 3b). 

Based on the abundance of the different bacterial functional groups, Chemo-1 was the most relevant metabolic potential associated with the carbon cycle in leaf and stem litter, especially in spring (Figure 3a,b). In addition to Chemo-1, the fermentation and Chemo-2 were more relevant in summer, and methylotrophy and oxygenic photoautotrophy were more relevant in autumn (Figure 3a,b). The abundance of bacterial metabolism potentials related to carbon cycle was the highest in leaf and stem litter in spring, compared with summer and autumn (Figure 4). Compared with the ambient temperature, constant and variable warming had a significantly promoted effect on photoheterotrophy metabolisms in leaf litter in spring, aromatic compound degradation in stem litter in spring and methylotrophy in leaf litter in autumn (*p* < 0.05, Figure 4a,b,e). However, constant and variable warming significantly inhibited fermentation and Chemo-2 in leaf and stem litter compared with the control during the summer season (*p* < 0.05, Figure 4c,d). Constant warming significantly enhanced photoheterotrophy metabolisms in stem litter in spring, while variable warming significantly increased aromatic compound degradation in leaf litter in spring and methylotrophy in stem litter in summer (*p* < 0.05, Figure 4a,b,d).

The functional groups in leaf litter of nitrogen cycle were nitrogen fixation (41% ± 0.09), ureolysis (35% ± 0.07), nitrate reduction (21% ± 0.04) and ammonification (3% ± 0.02) (Figure 3c). The functional groups in stem litter of nitrogen cycle were nitrogen fixation (32% ± 0.08), ureolysis (38% ± 0.09), nitrate reduction (23% ± 0.04) and ammonification (6% ± 0.03) (Figure 3d). Based on the abundance of the different bacterial functional groups, the nitrogen fixation, ureolysis and nitrate reduction were the most relevant metabolic potential associated with the nitrogen cycle in leaf and stem litter during the experiment period.

With regard to nitrogen cycle, the abundance of bacterial functional groups related to nitrogen cycle was the highest in summer compared with spring and autumn, and the abundance of bacterial function in the stem litter was much higher than that in the leaf litter (Figure 5). Constant and variable warming cause the significant inhibition of ammonification in leaf litter compared with the control in summer, while variable warming significantly promoted nitrogen fixation function in summer and ureolysis function in autumn (*p* < 0.05, Figure 5c,e). In stem litter, constant and variable warming significantly decreased nitrate reduction and ammonification functions in summer, while nitrogen fixation function in summer and ureolysis function in autumn significantly increased under variable warming treatment, and nitrogen fixation in autumn significantly enhanced under constant warming treatment, compared with the control (*p* < 0.05, Figure 5d,f). 

### 3.4. Bacterial Metabolic Potential and Environmental Parameters Analysis

Bacterial metabolic potential and environmental parameters driven by warming related to carbon and nitrogen cycle in leaf and stem litter showed potential consistency (Appendix A). There was a strong correlation between the bacterial metabolic potential related to carbon cycle and DO, conductivity, pH and ammonia nitrogen in leaf and stem litter (Figure 6 and Appendix A). DO and pH were significantly positively correlated with Chemo-1 and significantly negatively correlated with oxygenic photoautotrophy (*p* < 0.05, Figure 6a,b). Chemo-1 was significantly negatively correlated with conductivity and ammonia nitrogen (*p* < 0.05, Figure 6a,b). With regard to nitrogen cycle, nitrogen fixation was significantly negatively correlated with DO and positively correlated with conductivity and phosphorus (*p* < 0.05, Figure 6c,d). Different from leaf litter, there was a significant positive correlation between nitrogen fixation and ammonia nitrogen, ureolysis and DO, and phosphorus in stem litter (*p* < 0.05, Figure 6c,d).

## 4. Discussion

The central role and global importance played by microbial metabolic potential in climate change have recently been emphasized [52,53]. Microorganisms can affect climate change (such as the production and consumption of greenhouse gases) and can also be altered by climate change. In this study, we provided annotations for bacterial metabolic potential in different tissues of macrophyte litters through FAPROTAX data prediction using 16S high-throughput MiSeq sequencing data and clarified microbial metabolic potential in response to warming alters the decomposition process of aquatic plant litter. This study represents a first attempt to explore the functional patterns of microbial communities in different tissues of submerged macrophyte litter in shallow lakes under climate warming and guide future hypotheses for the study of shallow lakes under stress from a changing climate from a microbial perspective.

### 4.1. Bacterial Metabolic Potentials in Leaf and Stem Litter Response to Climate Warming during Decomposition

Various bacterial metabolic potentials respond differently to warming. Phototrophy and chemoheterotrophy are the main pathways of carbon flux within lakes communities [54]. As might be expected from the physical and chemical features of the mesocosms during the experiment, a prominent role of the aerobic chemoheterotrophy metabolism in litters was found during decomposition. Under warming treatments, the functional groups of aerobic chemoheterotrophy metabolism decreased in spring and summer but increased in autumn, which agrees with their nutrition conditions and the oxygen produced in the mesocosms. Heterotrophic metabolism may also be limited by light, as we found that photoheterotrophic metabolic potential is abundant, especially in spring when they are significantly promoted by warming. Photoheterotrophs (rhodopsin-containing bacteria and aerobic anoxygenic phototrophs) harness solar energy to produce ATP but cover most of their energy requirements through respiration, not fixing inorganic [55]. This may be related to changes in bacterial communities of proteobacteria and actinobacteria in mesocosms. It has been reported that there are proteobacteria and actinobacteria that can perform photoheterotrophication in freshwater ecosystems [56,57].

Warming leads to seasonal variation of DO concentration, which may explain the occurrence of microorganisms with potential anaerobic metabolism, such as Chemo-2 and methylotrophy. Chemo-2 contains the heterotrophic decomposition pathways of carbon compounds such as xylan and cellulose, which are important components of plant cell walls. In summer, constant and variable warming significantly reduced the Chemo-2 metabolism in stem and leaf litter. The significant change of xylan-degrading enzymes was related to temperature and pH [58]. Facultative methylotrophy was found in members of proteobacteria, actinomycetes and firmicutes [59]. Methylotrophy is an ancient metabolic feature [60], and hypoxic diffusion promotes anaerobic microbial metabolism [61,62]. Compared with spring, the oxygen content in the mesocosm is reduced in summer and autumn, promoting bacterial metabolisms such as fermentation and methylotrophy. There was an obvious upward trend of methylotrophy metabolism under warming treatments. Methylotrophy metabolism was significantly enhanced under variable warming in stem litter in summer and leaf litter under constant and variable warming in autumn. Methylotrophs are capable of growth on methanol as well as other reduced methylated one-carbon (C1) compounds such as methane, methylamine or formate [63,64] and may be greatly affected by changes in environmental conditions. For example, warming may promote oxidation reduction potential in water, thus accelerating methylotrophic metabolism [65]. The increase in nitrate content during the experiment also drives the anaerobic methane oxidation process [66]. Increased methylotrophic metabolism caused by warming promotes the utilization of reductively methylated one-carbon substrates by microorganisms, which may alter methane fluxes at the sediment–water interface.

Nitrogen fixation was the dominant bacterial metabolism potential associated with nitrogen cycle in leaf and stem litter. It has been found that bacterial groups using active carbon sources seem to be more inclined to fix nitrogen [67,68]. We found that nitrogen fixation metabolism potential increased under warming, especially in litter under variable warming in summer and stem litter under constant warming in autumn. The optimum temperature for nitrogenase is generally considered to be around 25 ℃, but it can be as high as 42 ℃ [69,70]. In studies of land plants, nitrogen fixation is reduced in cooler soils during most of the growing season [69]. In general, enzyme dynamics and metabolic activity are slowed down at low temperatures [71]. Therefore, warming may stimulate the activity of nitrogenase and enhance the nitrogen fixation metabolism potential, which also explains why nitrogen fixation metabolism potential was significantly enhanced in summer. Nitrogen fixation metabolism potential may be related to Alphaproteobacteria, which was found to be the most dominant bacterial community in litter in our previous study [21]. 

In addition to nitrogen fixation, ureolysis, nitrate reduction and ammonification were also bacterial metabolic potential communities with high abundance related to nitrogen cycle during our experiment. Ureolysis was the highest functional abundance in spring and increased significantly in autumn under variable warming. Nitrate reduction includes the assimilation and dissimilation process. In summer, dissimilated nitrate reduction mainly occurs in the anoxic or hypoxic conditions. Ammonification is the process in which nitrate is dissimilar to ammonium (DNRA). It was found that hypoxia enhanced both dissimilated nitrate reduction to nitrite and DNRA, suggesting that the two pathways may be mediated by the same microorganisms that perform the dissimilated pathways [72]. The bacterial functional abundance of nitrate reduction and ammonification in stem litter was significantly reduced under warming in summer. In contrast to our study, hybridization analysis based on GeoChip shows a significant increase in the abundance of genes involved in denitrification [73,74]. Warming can promote the degradation of recalcitrant carbon into unstable and dissolved organic carbon, and this additional electron donor supply can then stimulate heterotrophic denitrifiers [73,75]. However, a single temperature condition cannot explain the complex metabolic potential changes in the outdoor mesocosms, and the changes of bacterial function related to nitrogen cycle may be the result of the interaction of temperature and various environmental factors, such as inorganic nitrogen content, pH, composition of microbial communities, etc. [76,77]. The specific effects of warming on carbon and nitrogen cycle need to be further studied, and future studies could consider combining isotope and molecular techniques to elucidate the relative effects of physicochemical and biological factors [78,79].

### 4.2. Relationship between Bacterial Metabolism Potential and Environmental Factors under Climate Warming

Our redundancy analysis of environmental factors of litter found that DO, pH, conductivity and ammonium nitrogen were the main environmental factor affecting bacterial metabolic potential related to carbon and nitrogen cycle. However, environmental factors could not fully explain the difference in bacterial metabolic potential (the proportion of explanation for the first two axes was only 37.96–43.68%), further indicating the important role of habitat microenvironment in bacterial community construction. 

The correlation between main bacterial functional communities and environmental factors heat map further analyzed the response of different bacterial metabolic potential to environmental factors. Changes in environmental parameters driven by warming indirectly affect the metabolic processes of different bacterial metabolic potential. Firstly, aerobic chemoheterotrophy showed a strong positive correlation with DO and pH (|r| > 0.75) in our experiment. Therefore, the increase in DO concentration may promote the heterotrophic metabolic potential of bacteria. The low pH value of soil will inhibit the enzymatic metabolic activities of bacteria, which is not conducive to the growth of bacteria [80,81]. Secondly, increased soil pH can promote bacterial diversity by releasing dissolved organic matter [82]. In addition, soil pH can indirectly affect bacterial diversity through other covariant factors, such as nutrient availability [83]. 

Among the bacterial metabolic potentials related to nitrogen cycle, only nitrogen fixation metabolic potential has a significant relationship with environmental factors. Nitrogen fixation is usually a significant source of nitrogen in a nitrogen-restricted environment [84]. Our results showed that warming promoted the increase in inorganic nitrogen and phosphorus contents, and there was a significantly positive correlation between nitrogen fixation functional group. Previous studies have confirmed that the availability of inorganic nitrogen and phosphorus and temperature limit the development of nodule formation and the activity of nitrogenase in nitrogen-fixing plants, and it has been used to explain the reasons affecting nitrogen fixation of nitrogen fixing plants [69,70,85,86]. Therefore, the enhancement of the nitrogen fixation function may be the result of both water temperature elevated and inorganic nitrogen and phosphorus content increasing under warming. In addition, the DO content and pH were negatively correlated with the nitrogen fixation functional group. Since nitrogenase is irreversibly inhibited by oxygen, the cell requires lower conditions or increased respiration to free the cell of oxygen and thus achieve nitrogen fixation [87,88]. Studies on legumes found that the number of nodule and nitrogen fixation ability increased under acidic conditions, but nitrogenase activity decreased sharply at pH 7.0 [89]. In studies on sediment nitrogen fixation, nitrogen fixation rate is significantly negatively correlated with sediment pH [90]. Therefore, pH may influence sediment nitrogen fixation by regulating metabolism in nitrogen fixation prokaryotes [91]. 

### 4.3. Differences in the Responses of Bacterial Metabolic Potential of Leaf and Stem Litter to Warming and Environmental Factors

Most of the bacterial functional groups in the leaf and stem litter of *P. crispus* showed no significant difference under warming. In our previous study, it was found that there was no significant difference in decomposition rates and decomposition-related microbial communities of litter from different tissues [21]. However, we still found differences in the part of bacterial metabolic potential in leaf and stem litter. For example, the potential for fermentation of elementary carbon in leaf litter significantly decreased under warming in summer, but no difference was shown in stem litter. Furthermore, the abundance of bacterial metabolic potential related to the nitrogen cycle in stem litter was higher than in leaf litter, such as nitrate reduction and ammonification in summer. The main reason for this may be the differences in quality of litter (i.e., initial lignin contents, N content and C:N ratios) in different plant tissues [92,93,94]. Furthermore, it was found that the changes in temperature and other environmental factors could cause the changes of specific elements in plant tissues [95,96]. Under climate warming, the content of carbon and nitrogen elements in plants and their distribution characteristics in different organs and tissues will also have adaptive changes [97], which will also change their palatability to upper level consumers, resulting in the change of residual matrix components in leaf and stem litter, thus affecting the change of bacterial metabolic potential [95,97]. The heatmap of bacterial metabolic potential and environmental factors showed that the bacterial metabolic potential in the stem litter was more strongly responsive to environmental factors. For example, ureolysis capability had a significant positive correlation with DO and pH in the stem litter, but no significant correlation in the leaf litter. This may also be attributed to the differences in the quality of litter between leaf and stem. At present, most studies on the decomposition of litter have selected land plants as materials to discuss the decomposition of leaf litters of different species of plants [98,99]. For example, studies on different litter types in terrestrial ecosystems found that the decomposition rate of broad-leaved tree species was significantly higher than that of coniferous tree species [100]. Coniferous forests have lower nutrients (such as nitrogen and phosphorus) and higher structural components (such as lignin and cellulose) and recalcitrant compounds (such as tannins and polyphenols) than broadleaved forests, and therefore decompose more slowly [100,101]. Therefore, considering the joint effect of warming and litter traits aspects allow a more refined understanding of the underlying mechanisms of climate change effect on ecosystem functioning [7,102].

## 5. Conclusions

Our results successfully reflect the response of bacterial functional groups to different warming patterns and environmental factors, which should have an important implication on biogeochemical cycles in aquatic environment. Our results suggest that warming shifted the functional structure of bacterial communities in leaf and stem litter during decomposition. In litter, the metabolic potential of bacteria was mainly related to the carbon cycle and was mainly heterotrophic decomposition. Warming leads to the decrease in main chemoheterotrophic metabolic potential. Notably, increased methylotrophic metabolism caused by warming promotes the utilization of reductively methylated one-carbon substrates by microorganisms, which means that further warming may alter methane-cycling microbial metabolism. The bacterial metabolic potential of nitrogen cycle was abundant in summer, and nitrate reduction was significantly inhibited by water temperature rising, while nitrogen fixation significantly increased under variable warming. The main environmental factors driven by warming were DO, pH, conductivity and ammonium nitrogen, which affect the bacterial metabolic potential related to carbon and nitrogen cycle. The effects of warming and environmental factors on fermentation, nitrate reduction and ammonification capabilities in stem and leaf litter were different, and the bacterial metabolic potential of stem litter was more strongly responsive to environmental factors. These observations may help to recognize the potential importance of microbial functional responses to climate change in the mesocosms. Further studies should access the joint effect of warming and litter trait on decomposition and reveal major microbial players and their biogeochemical functions.

## Figures and Tables

**Figure 1 microorganisms-10-01327-f001:**
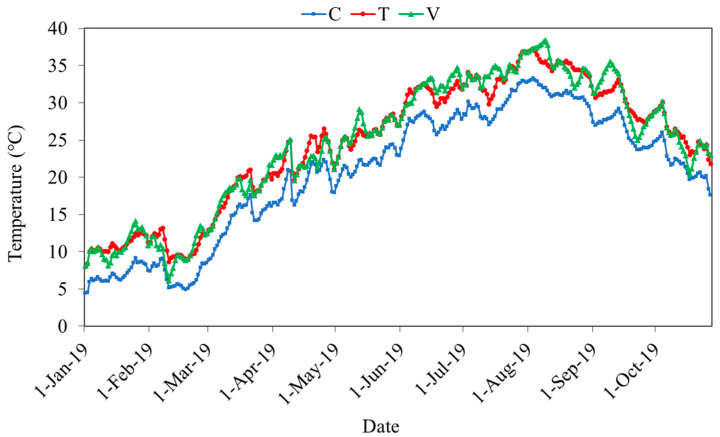
Water temperature for all treatments during the experiment. C, ambient temperature; T, constant warming; V, variable warming.

**Figure 2 microorganisms-10-01327-f002:**
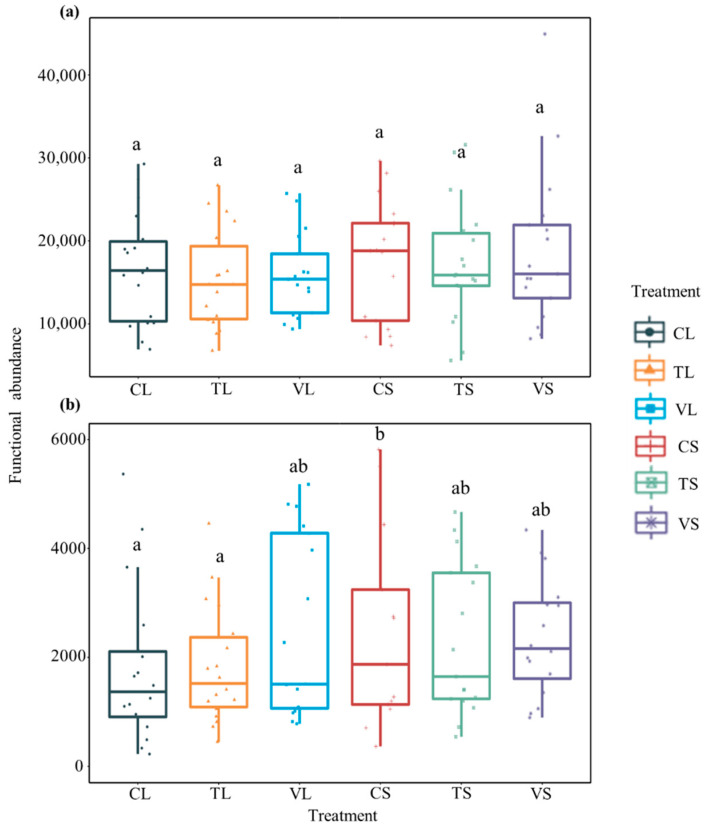
Abundance of bacteria related to C cycle (**a**) and N cycle (**b**) under different treatments in leaf and stem litter. C, ambient temperature; T, +4 °C constant warming; V, variable warming; L, leaf litter; S, stem litter. All data are presented as mean ± SE. The letters above the bars represent significant differences between treatments (*p* < 0.05, Tukey’s test).

**Figure 3 microorganisms-10-01327-f003:**
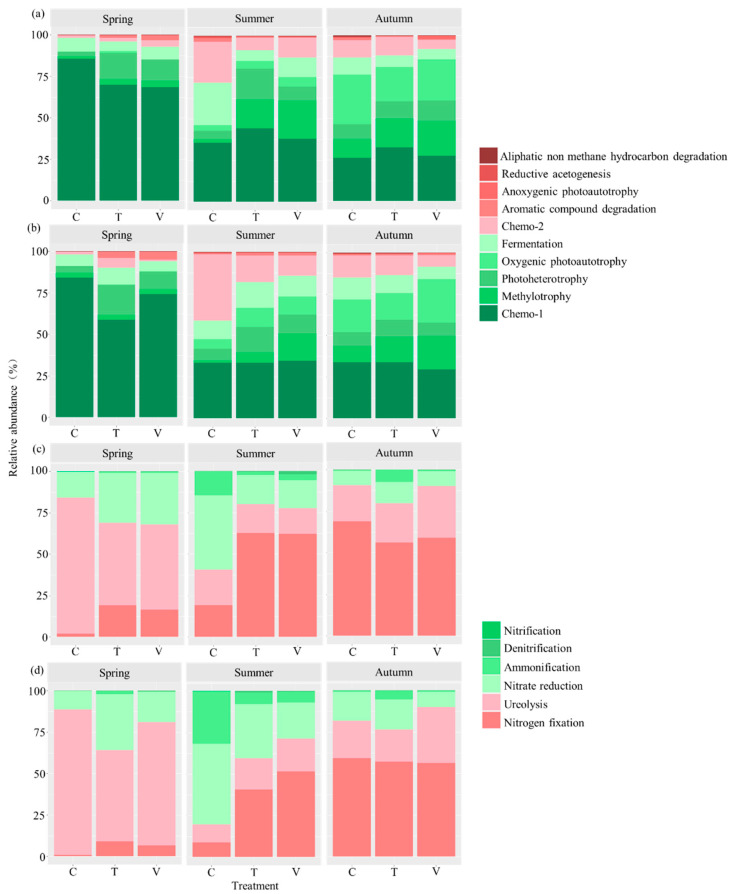
The relative functional abundance of bacteria related to carbon cycle in leaf (**a**) and stem litter (**b**), and the relative functional abundance of bacteria related to the nitrogen cycle in leaf (**c**) and stem litter (**d**).

**Figure 4 microorganisms-10-01327-f004:**
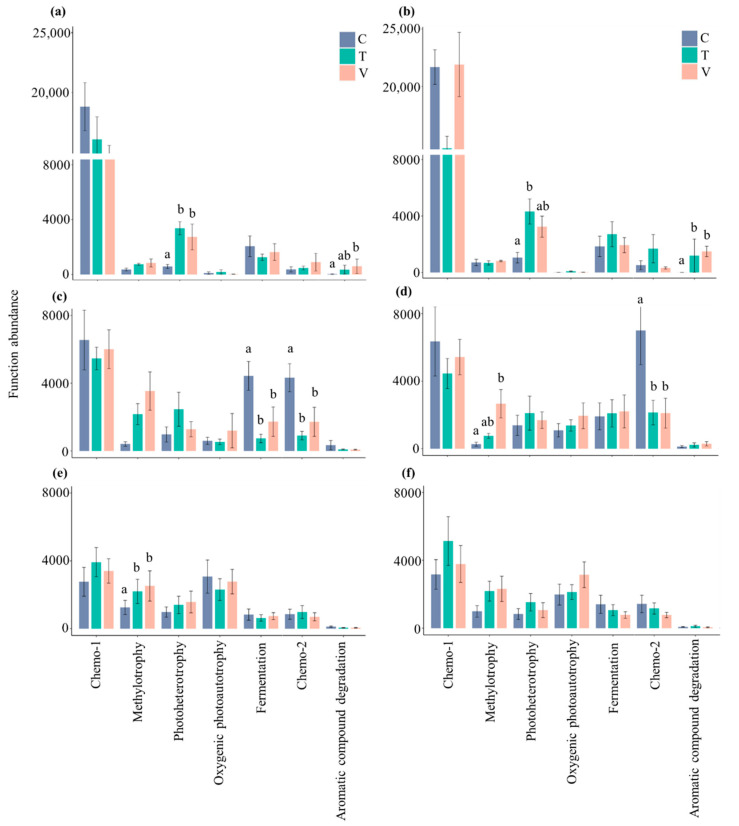
The columnar difference in bacterial functional groups related to the carbon cycle in leaf and stem litter. (**a**), leaf litter in spring; (**b**), stem litter in spring; (**c**), leaf litter in summer; (**d**), stem litter in summer; (**e**), leaf litter in autumn; (**f**), stem litter in autumn. All data are presented as the mean ± SE. Significant (*p* < 0.05, Tukey’s test) differences among treatments are indicated by letters above the bars. The figure shows the more than 95% of the abundance of bacteria functional groups.

**Figure 5 microorganisms-10-01327-f005:**
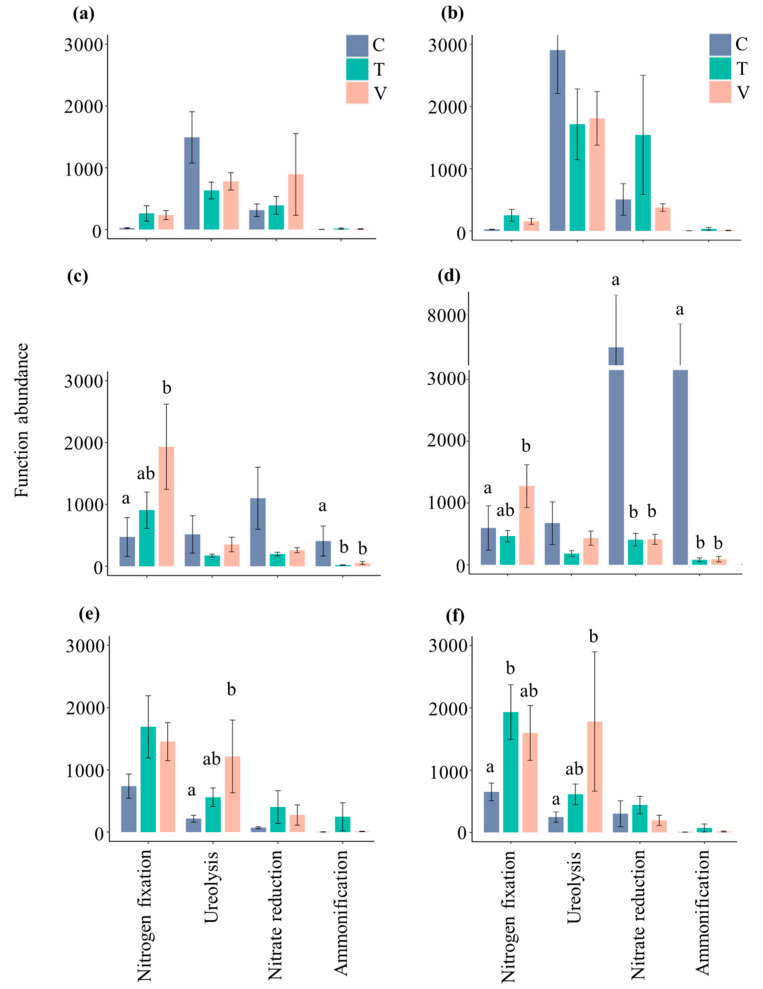
The columnar difference in bacterial functional groups related to the nitrogen cycle in leaf and stem litter. (**a**), leaf litter in spring; (**b**), stem litter in spring; (**c**), leaf litter in summer; (**d**), stem litter in summer; (**e**), leaf litter in autumn; (**f**), stem litter in autumn. All data are presented as the mean ± SE. Significant (*p* < 0.05, Tukey’s test) differences among treatments are indicated by letters above the bars. The figure shows the more than 95% of the abundance of bacteria functional groups.

**Figure 6 microorganisms-10-01327-f006:**
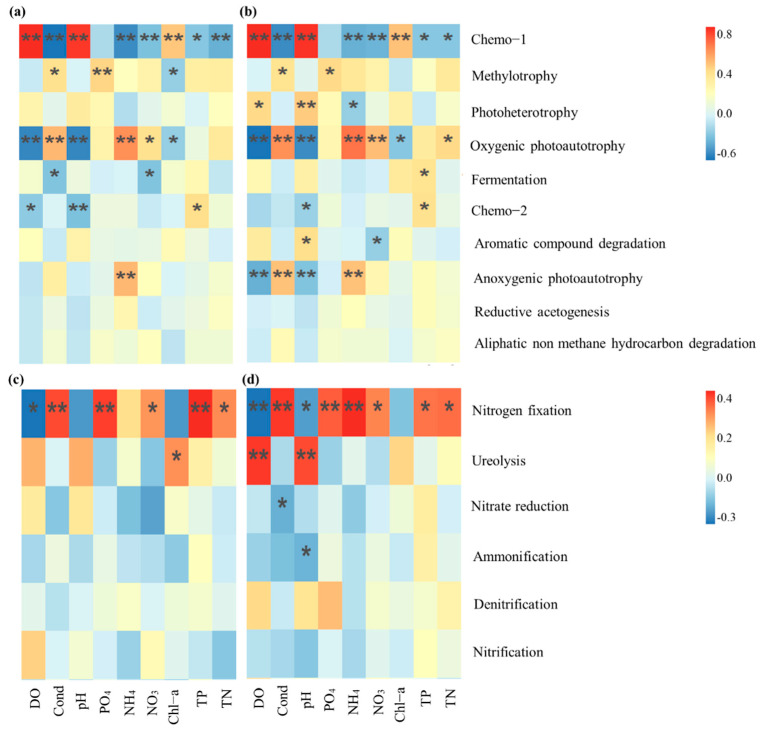
Heatmap correlation between relative abundance bacteria functional groups related to carbon and nitrogen cycles in leaf and stem litter and environmental factors. (**a**), leaf litter (related to carbon cycle); (**b**), stem litter (related to carbon cycle); (**c**), leaf litter (related to nitrogen cycle); (**d**), stem litter (related to nitrogen cycle). Level of significance at *p* < 0.05 and *p* < 0.01 is denoted by * and **, respectively.

**Table 1 microorganisms-10-01327-t001:** Mean water quality for all treatments during the experiment. Values are mean ± SE. C, ambient temperature; T, +4 ℃ constant warming; V, variable warming. The letters above the data represent significant differences between treatments (*p* < 0.05, Tukey’s test).

		pH	Cond (μs·cm^−1^)	DO (mg·L^−1^)	TN (mg·L^−1^)	TP (μg·L^−1^)	PO_4_^3−^-P (μg·L^−1^)	NH_4_^+^-N (mg·L^−1^)	NO_3_^−^-N (mg·L^−1^)	Chl-a (μg·L^−1^)
Spring	C	9.43 ± 0.18 ^a^	170.57 ± 7.81 ^a^	12.44 ± 0.44 ^a^	0.38 ± 0.04 ^a^	0.03 ± 0.01 ^a^	5.81 ± 1.33 ^a^	0.10 ± 0.01 ^a^	0.08 ± 0.01 ^a^	11.11 ± 1.47 ^a^
	T	9.43 ± 0.18 ^a^	230.48 ± 7.53 ^b^	10.51 ± 0.40 ^a^	0.67 ± 0.06 ^b^	0.07 ± 0.02 ^a^	17.91 ± 7.91 ^a^	0.11 ± 0.02 ^a^	0.12 ± 0.02 ^a^	14.01 ± 2.33 ^a^
	V	9.49 ± 0.12 ^a^	212.31 ± 7.16 ^b^	10.39 ± 0.56 ^a^	0.54 ± 0.07 ^ab^	0.06 ± 0.01 ^a^	26.32 ± 7.77 ^a^	0.16 ± 0.02 ^a^	0.11 ± 0.02 ^a^	15.67 ± 2.85 ^a^
Summer	C	7.59 ± 0.17 ^a^	180.05 ± 9.44 ^a^	3.74 ± 0.74 ^a^	0.65 ± 0.08 ^a^	0.08 ± 0.02 ^a^	58.38 ± 34.24 ^a^	0.09 ± 0.02 ^a^	0.11 ± 0.01 ^a^	4.77 ± 1.09 ^a^
	T	8.31 ± 0.18 ^b^	247.06 ± 11.10 ^b^	5.92 ± 0.52 ^b^	1.03 ± 0.11 ^b^	0.13 ± 0.03 ^a^	59.45 ± 9.30 ^ab^	0.19 ± 0.04 ^a^	0.18 ± 0.03 ^b^	11.12 ± 3.67 ^a^
	V	8.03 ± 0.19 ^ab^	243.77 ± 6.39 ^b^	4.19 ± 0.50 ^ab^	0.81 ± 0.05 ^ab^	0.12 ± 0.02 ^a^	98.50 ± 24.36 ^b^	0.17 ± 0.02 ^a^	0.15 ± 0.01 ^ab^	3.30 ± 0.51 ^a^
Autumn	C	7.85 ± 0.19 ^a^	265.91 ± 18.40 ^a^	4.27 ± 0.98 ^a^	0.58 ± 0.07 ^a^	0.04 ± 0.00 ^a^	20.21 ± 3.68 ^a^	0.21 ± 0.04 ^a^	0.18 ± 0.01 ^a^	1.93 ± 0.32 ^a^
	T	8.38 ± 0.16 ^a^	315.61 ± 11.81 ^b^	6.85 ± 1.12 ^a^	1.25 ± 0.11 ^b^	0.06 ± 0.01 ^a^	58.05 ± 25.31 ^a^	0.42 ± 0.17 ^a^	0.23 ± 0.03 ^a^	2.57 ± 0.32 ^a^
	V	8.25 ± 0.15 ^a^	335.50 ± 9.83 ^b^	5.55 ± 0.91 ^a^	1.02 ± 0.09 ^b^	0.05 ± 0.01 ^a^	22.62 ± 4.17 ^a^	0.28 ± 0.04 ^a^	0.23 ± 0.01 ^a^	2.17 ± 0.41 ^a^

## Data Availability

Our paired-end Illumina sequence data used in summer are available in the Sequence Read Archive (SRA) of the National Center for Biotechnology Information (NCBI), BioProject accession number PRJNA705584. The paired-end Illumina sequence data in spring and autumn have not been published.

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
