# Peer review of "Bacterial Metabolic Potential in Response to Climate Warming Alters the Decomposition Process of Aquatic Plant Litter—In Shallow Lake Mesocosms"

_microorganisms, 2022, doi:10.3390/microorganisms10071327_

Round 1

Reviewer 1 Report

Lakes are critical natural resources that are sensitive to changes in climate which may alter the composition and function of lake bacterial communities. The manuscript continues the publications of the author's team devoted to the warming impact on microbial processes in the mesotrophic shallow lake under controlled conditions of the mesocosms. Earlier, according to the same scheme, the authors investigated changes in the bacterial metabolic function in the lake water and sediment (Shi et al., Water 2022, 14, 1203)  and   microbial communities composition of the typical submerged macrophyte (Potamogeton crispus L) litter. (Pan et al. Water 2021, 13).

The problem is actual and important, however, there are some concerns with the description of experimental design and results, and this means I cannot approve the manuscript in the current form. Below I will discuss my main comments and suggestions, which hopefully can help the author/s improve the study.

The greatest concern is a description of the experimental approaches. Firstly, as described in Methods (L. 147-148), the bags for Illumina analysis were collected on the 30th day, but results for spring, summer, autumn are discussed. Why do only summer Illumina sequence data has been published? Then, section 2.4 is more results, than methods, but there is no exact description how  the abundance of bacteria with specific metabolic functions was evaluated.

Another point is incorrect expressions when describing results. I think the term “functional groups on leaves (stems)” is incorrect, because the bacterial communities originated from the lake sediment and participated in litter decomposition were analyzed. It will be reasonable to compare results from MS with the data in Pan et al 2021. I can´t agree with the terms “total abundance of bacterial metabolic function”, “metabolic function total abundance”, “ the functional groups in stem litter of nitrogen cycle”, etc.

Minor points:

1.        L.82 – “microbial community classification” – what does it mean?

2.       “litters” (L. 62, 69, 73 , etc) I think it would be better to use this word in the singular.

3.       L.88 – unclear sentence.

4.       L. 89 - the sentence is not finished

5.       The hypothesis needs to be reformulated and concretized.

6.       What does Fig 3 demonstrate?

Reviewer 2 Report

The manuscript by Shi et al. reports data to test their hypotheses that climate warming changes the bacterial functional structure of litters in shallow lake mesocosms. The study is well planned and conducted, and the results are interesting. Yet the authors over-interpret their data, building their entire analyses on predicted metabolic pathway genes derived from partial 16S sequences (Illumina). The conclusions go further than the data used can support.

Metabolic pathway prediction derived from previously sequenced genomes selected by 16S sequence data matches is powerful. Yet these derived data must be treated as predictions of potential, rather than actual data, and that for at least two reasons:

1: Not all strains of an OUT (species) contain the same genes. In fact, the pangenomes of widely sequenced species are large, so that each strain contains only a sub-set of the genes. This means that some strains lack functions found in the sequenced genomes, while others contain additional sets of genes. Furthermore, Illumina sequencing generally yields only the partial 16S sequence (e.g. V3-4), so that sequenced are generally matched at the genus rather than species level, placing further levels of uncertainty on predicted metabolic function. A case in point here is nitrogen fixation. I am not aware of a single bacterial genus where are species and their strains contain the core nitrogenase genes – even in the rhizobia.

2: The majority pf bacterial genes are regulated, meaning that gene presence is not hard evidence of the expressed protein, so gene-derived functions can merely be viewed as potential. A case in point here is nitrogen fixation as expression of nifHDK is tightly regulated.

As example, the paragraph starting line 357 states that “… was the dominant bacterial metabolic function…”. This statement is too strong because a) there are no known cases where all members of a genus contain nifHDH genes, and b) nitrogenase is tightly regulated. The authors are requested to rephrase all statements that imply gene expression as fact and rephrase as something such as “metabolic potential”, or “there is a probability that XYZ is expressed”.

The overall experimental design is outlined clearly, but there is too little information on sample types and time points of biomass sampling for DNA extraction and 16S PCR in the methods section. One has to piece this information together from the results. Also, the authors are requested to define the specific 16S region targeted, and what primer set was used. Furthermore, the authors are asked to clarify whether the 16S samples were sequenced separately for the 6 reps (18 mesocosms, two treatments and one control so I assume 6 each), or whether the DNA extracts / PCR amplicons were pooled before sequencing. If these were sequenced separately, then the data should be subjected to multivariate analysis such as for example NMDS.  This would be valuable to show variation among reps.

Specific points for attention:

1.     Ln 43:  “The shallow lakes” – surely this is not universally true but linked to lakes in certain geographical or climatic regions.

2.     Ln 44: “growth and decline” of what?

3.     Ln 65-66: Incomplete sentence. Suggestion: “ …rates for wood were observed / reported [19] …”

4.     Ln 89-90: Incomplete sentence. Suggestion: “…during decomposition is unknown.”

5.     Ln 94: “change” to changes

6.     Ln 98: “mesocosm” to mesocosms

7.     Ln 122: there were only two treatments as (1) was the control.

8.     Section 2.4: The section begins with Fnprotax, but there is no information on times and types of samples taken, or how they were processed (DNA extraction and PCR).

9.     Ln 210: Which statistical test was used to determined “significance”?

10.  Ln 370: Rhizobia are not the predominant part of the alfaproteobacteria.

11.  Ln 468: The line “First study reporting functional metabolism…” is too strong of a conclusion based on the data available here.

Round 2

Reviewer 1 Report

The authors significantly revised the text of the MS and mainly took into account the comments of the reviewer. At the same time, there are minor stylistic flaws in the text. Perhaps this is due to the fact that the authors are not native English speakers.

In the MS the relative abundance of the bacteria  that potentially have metabolic functions related to  carbon and nitrogen cycling is discussed. Therefore, I think it is appropriate to use the terms "bacterial functional groups", 'abundance of bacteria related to C (N) cycle", maybe " bacterial metabolic potential" in data discussion. But I can't agree with "total abundance of bacterial metabolic potential" (L.286), "Metabolic potential  total abundance of bacteria" (L .292), "3.3. Composition of bacterial functional community" (L. 298), etc. Please make corrections throughout the MS text.

Reviewer 2 Report

Thank you for your revised manuscript and attending to all the comments made.
